# Reliability, Validity and Responsiveness of the Polish Version of the Western Ontario Shoulder Instability Index (WOSI-PL) in Patients after Arthroscopic Repair for Shoulder Instability

**DOI:** 10.3390/ijerph192114015

**Published:** 2022-10-27

**Authors:** Agnieszka Bejer, Jędrzej Płocki, Marek Kulczyk, Sharon Griffin, Ireneusz Kotela, Andrzej Kotela

**Affiliations:** 1Institute of Health Sciences, College of Medical Sciences, University of Rzeszow, Rejtana 16C, 35-959 Rzeszow, Poland; 2The Holy Family Specialist Hospital, Rudna Mała 600, 36-060 Głogów Małopolski, Poland; 3Department of Physiotherapy, Collegium Medicum, University of Information Technology and Management in Rzeszow, Sucharskiego 2, 35-225 Rzeszow, Poland; 4Fowler Kennedy Sport Medicine Clinic, Faculty of Medicine, University of Western Ontario, 3M Centre, 1151 Richmond Street, London, ON N6A 3K7, Canada; 5Institute of Health Sciences, Collegium Medicum, The Jan Kochanowski University, IX Wieków Kielc 19a, 25-317 Kielce, Poland; 6Department of Orthopaedic Surgery and Traumatology, Central Clinical Hospital of the Ministry of Interior, Wołoska 137, 02-507 Warsaw, Poland; 7Faculty of Medicine, Collegium Medicum, Cardinal Stefan Wyszyński University, Woycickiego 1/3, 01-938 Warsaw, Poland

**Keywords:** shoulder instability, validation, psychometrics, questionnaire, WOSI, arthroscopy, physiotherapy

## Abstract

Apart from imaging and physical examination for shoulder instability (SI), medical history with patient feedback should be considered to assess the patient’s condition and recovery. The aim of this study was to evaluate psychometric properties of the Polish version of Western Ontario Shoulder Instability Index (WOSI)—one of the most frequently used patient-reported outcome measures for SI. During examination 1, 74 patients after arthroscopic repair for SI (age $\bar{x}$ = 30.01 ± 8.98) were tested. Examinations 2 and 3 involved 71 and 51 patients, respectively. They completed the Polish version of the WOSI, the shortened version of the Disabilities of Arm, Shoulder and Hand Questionnaire (QuickDASH), the Short Form-36 version 2.0 (SF-36 v. 2.0) and 7-point Global Rating Change scale (GRC). The WOSI-PL demonstrated high internal consistency (Cronbach’s alpha for total = 0.94), and test–retest reliability (Total ICC_2,1_ = 0.99). High construct validity was found (89%) as the a priori hypotheses were confirmed. All domains and total scores of WOSI-PL showed a moderate to strong degree of responsiveness (ES = 0.37–0.44; SMR = 0.87–1.26). Minimal clinically important difference (MCID) for the Total WOSI-PL was 126.43 points/6% (95%CI 67.83–185.03) by the anchor-based method and 174.05 points/8% (95%CI 138.61–233.98) by the distribution-based method. The Polish version of the WOSI can be considered a reliable, valid and responsive PROM. It is recommended for assessing the quality of life in patients after arthroscopic repair for SI and can be applied in research and in the clinical setting for monitoring treatment and facilitating patient-centred therapeutic decisions.

## 1. Introduction

Shoulder instability arises from traumatic or atraumatic causes. Its classification includes anterior, posterior or multidirectional instability. Pathologic conditions that may contribute to instability in the glenohumeral joint include joint laxity, labral tears, ligament injuries, impaired muscular control as well as bone defects within the humeral head or in the glenoid [1,2]. Shoulder instability may cause pain, popping and crepitus, a sensation of the joint giving way, limb weakness, repeated episodes of joint dislocation, reduced participation in activities of daily living and sports, and a deterioration in quality of life. Deterioration of joint function may develop over time [2].

Apart from imaging and physical examinations of shoulder instability, medical history with patient feedback should be considered to assess the patient’s condition and recovery. In the orthopedic literature the most frequently described patient-reported outcome measures (PROMs) regarding shoulder instability are the Western Ontario Shoulder Instability Index (WOSI), the Rowe Score, the Constant Score and the Oxford Shoulder Instability Score (OSIS). All of these disease-specific questionnaires demonstrate appropriate validity, reliability and responsiveness for patients with shoulder instability [3,4,5,6].

The Western Ontario Shoulder Instability Index (WOSI), developed by Kirkley et al. in 1998, contains 21 items grouped in four domains concerning the symptoms of instability, everyday functioning and quality of life. The WOSI contains instruction for users clarifying every item [3,7]. High validity and reliability of the WOSI was demonstrated, with better responsiveness than other shoulder measurement tools—the Disabilities of the Arm, Shoulder and Hand tool (DASH), the American Shoulder and Elbow Surgeons Standardized Shoulder Assessment Form (ASES), UCLA Shoulder Rating Scale, Constant Score, Rowe Rating Scale and a global health instrument—the SF12 [3]. Rouleau et al., in their systematic review, demonstrated that out of 25 questionnaires used to assess shoulder instability, the WOSI has the best psychometric features [8]. Whittle et al., in their systematic review, also strongly recommend using WOSI for patients with shoulder instability [6]. To our knowledge, the WOSI has been translated to Swedish [9], German [10,11], Japanese [12], Italian [13], Brazilian, Portuguese [14], Dutch [15,16], French—validated on a Canadian and Swiss population [17], Spanish [18], Danish [19], French—validated on a French population [20], Turkish [21], Hebrew [22], Arabic [23,24] and European Portuguese [25]. In Poland, translation and cultural adaptation of the WOSI was conducted by Bejer et al., in 2019 [26]. It met the guidelines of MAPI Research Institute and had the involvement of the author of the source version [27]. This multi-staged study resulted in the creation of a well-translated and culturally adapted version of the Polish questionnaire [26].

The aim of this study was to evaluate psychometric properties—reliability, validity and responsiveness of the Polish version of the WOSI. This research was undertaken because no disease-specific instrument currently available in Poland enables evaluation of various aspects of functioning and quality of life in individuals with shoulder instability.

## 2. Materials and Methods

### 2.1. Ethics

The research project was approved by the institutional Bioethics Committee at the University of Rzeszow (Resolution No. 1/6/2017). The study was carried out in compliance with Polish legal regulations and ethical standards specified in the relevant version of the Declaration of Helsinki (64th WMA General Assembly, Fortaleza, Brazil, October 2013). All the study participants provided informed written consent.

### 2.2. Subjects

This prospective cohort study with a repeated-measures design was performed from 2019 to 2020. All patients >18 years who had been treated with arthroscopic repair for shoulder instability in the Specialist Hospital in Rudna Mała, Poland, whether anterior, posterior or multidirectional, traumatic or non-traumatic, and were native Polish speakers were approached for inclusion in the study. Those that gave informed consent were enrolled. Exclusion criteria included: a history of surgery on the shoulder or the upper limb, a diagnosis other than shoulder instability that would contribute to the patient’s shoulder dysfunction (i.e., primary shoulder arthritis, shoulder fracture, acromioclavicular joint abnormality and cervical spine disease), co-existing neoplastic, rheumatic or neurological conditions which could interfere with the patient’s quality of life, or inability to understand or complete the questionnaires due to psychiatric or cognitive disorders.

### 2.3. Measurements

#### 2.3.1. The Western Ontario Shoulder Instability Index (WOSI)

It includes 21 items and four domains, namely physical symptoms (10 items), sports/recreation/work (4 items), lifestyle (4 items), and emotional wellbeing (3 items). Each item is scored on a 100 mm visual analogue scale. Total score is in the range between 0 and 2100, with higher scores reflecting extreme limitations in shoulder-related quality of life. Additionally, the score can be expressed as a percentage of normal shoulder function, a score of 2100 corresponding to 0% of normal function and a score of 0 corresponding to 100% [3,7].

#### 2.3.2. The Shortened Version of the Disabilities of Arm, Shoulder and Hand Questionnaire (QuickDASH)

This self-administered questionnaire is intended to evaluate disability of the upper extremity. It consists of 11 questions: three related to symptoms and eight questions assessing the effects of upper limb problems reflected by limitations at work, in activities of daily living and in social activity. The response format is based on a 5-point Likert scale. The QuickDASH constraint and symptom index is determined by adding the circled digits, dividing the total by the number of responses and then subtracting 1 and multiplying by 25. The index is a number between 0 and 100, a higher value reflecting a greater limitation during activities [28]. In this study we have used the translated and validated Polish version of the QuickDASH questionnaire by Golicki et al. [29].

#### 2.3.3. Short Form-36 Version 2.0 (SF-36 v. 2.0)

This generic tool, designed to assess health-related quality of life (HRQOL), is comprised of 36 items representing eight domains: bodily pain, general health and mental health, vitality, physical functioning, social functioning as well as role limitations due to physical health and role limitations due to emotional health. The score in each domain ranges between 0 and 100, the lower score reflecting poorer quality of life [30]. Two separate summary scores, corresponding to physical health (PCS—Physical Component Summary) and mental health (MCS—Mental Component Summary), can be calculated based on these eight dimensions [31]. In this study we have used the translated and validated Polish version of the SF-36 questionnaire by Tylka et al. [32].

#### 2.3.4. Seven-Point Global Rating Change Scale (GRC)

The GRC scale was used to assess the patient’s improvement or deterioration over time in quantitative terms. Patients can report changes in their symptoms following a physiotherapy program, by rating them on a 7-point scale (1 = completely recovered, 2 = much better, 3 = better, 4 = slightly better, 5 = unchanged, 6 = slightly worse, 7 = very much worse) [33].

### 2.4. Study Procedure

The participants were assessed three times. The baseline examination (>12 to 16 weeks after surgery, Test 1) consisted of completing WOSI-PL, QuickDASH and SF-36 v. 2.0. During the second examination (Test 2; 48–72 h after Test 1), patients completed only WOSI-PL. The participants undergoing physiotherapy after Test 2 were included in Test 3, 8 weeks after the baseline examination, and they completed WOSI-PL and QuickDASH. They also reported symptom changes on a 7-point GRC.

### 2.5. Statistical Analyses

The statistical analyses were conducted using the Statistica 10.0. Normality of data was tested with the Shapiro–Wilk test. The level of statistical significance was set a priori at *p* ≤ 0.05.

#### 2.5.1. Sample Size

A post-hoc analysis of the test power was carried out for Intraclass Correlation Coefficient (ICC), for the null hypothesis ICC = 0.7 with a sample size of 71 subjects, a significance level of 0.05 and the expected ICC value in our population. The test power was found to be very high, exceeding 99.9% for each domain and the total score. The result showed that the sample size was satisfactory in terms of the statistics.

#### 2.5.2. Reliability Analysis

##### Internal Consistency

Cronbach‘s alpha coefficient (α) was applied to verify the internal consistency. A coefficient ranging from 0.70 to 0.95 was considered satisfactory [34,35].

##### Reliability (Test–Retest)

Assessment of the test–retest reliability was conducted using the intraclass correlation (ICC2,1), with a 95% confidence interval (CI). ICC equal to or greater than 0.70 was assumed to reflect positive rating of reliability [33,34].

##### Measurement Error (Test–Retest)

Assessment of error was performed using the standard error of measurement (SEM) and minimal detectable change at the 95% confidence level (MDC95) [36,37].

#### 2.5.3. Construct Validity Analysis

##### Tested Hypotheses

A priori hypotheses, which were developed to evaluate WOSI-PL in terms of the construct validity, covered the expected relationships between WOSI-PL and the comparison tools (depending on the construct similarity). The Spearman’s correlation coefficients (SCC) were computed for WOSI-PL Total and domain scores, and the region-specific QuickDASH (assessing upper extremity function) as well as SF-36 (i.e., a general quality of life questionnaire). The hypotheses were formulated separately by two authors (AB and JP), then the anticipated correlation agreement was assessed. Eighteen hypotheses were selected for the analyses. We hypothesized there would be strong negative correlations between WOSI-PL Total and the QuickDASH (1), and moderate positive correlations between the WOSI-PL total and the SF-36 (PCS, MCS, domains) (10). We also expected that there would be stronger correlations between the WOSI-PL and the QuickDASH than between the WOSI-PL and the SF-36 MCS and PCS (2). Furthermore, we anticipated that the WOSI-PL Total would correlate more significantly with SF-36 PCS than with MCS (1), and that there would be a stronger correlation between WOSI-PL domains—Physical symptoms, Lifestyle, Sports/recreation/work and SF-36 PCS compared to MCS (3), and, conversely, we anticipated there would be a more significant correlation between emotional domain of WOSI-PL and SF-36 MCS compared to PCS (1). It was determined that SCC r strength for validity corresponded to the following values: ≤0.30 = weak, 0.3 < r < 0.7 = moderate and >0.70 = strong. Construct validity of the WOSI-PL was considered high, or moderate or low, if the hypotheses were rejected at the rates: lower than 25%, in the range of 25–50% or more than 50%, respectively [34].

#### 2.5.4. Responsiveness

Significance of changes in WOSI-PL, observed between Test 1 and 3, was assessed using the Wilcoxon test for paired samples.

The standard effect size (ES) and standardized response mean (SRM):

ES was defined as a change in WOSI-PL score (Test 1 versus Test 3) divided by baseline SD, whereas SRM was computed from the mean score change divided by SD of the same score change. Absolute values below or equal to 0.20, in the range of 0.21–0.79, and equal to or over 0.80 represent low, moderate, and high responsiveness, respectively, for ES and SRM [38].

The correlation of changes in WOSI-PL (Test 1–3) and those in QuickDASH (Test 1–3) was assessed using Spearman correlation coefficient.

The minimal clinically important difference (MCID):

The MCID was calculated based on two methods: anchor-based method (mean change in the group “with change”) and distribution-based method (SEM for patients “without change”) [39]. In both methods the group “without change” was assumed for patients who marked a GRC score equal to 4, 5 or 6, and the group “with change” for the patients with the scores of 1, 2 or 3.

## 3. Results

### 3.1. Participant Characteristics

Following arthroscopic repair for shoulder instability, 74 subjects completed questionnaires during Test 1. Patient gender, age, operated limb, hand dominance and time from surgery were recorded (Table 1). For the purpose of assessing test–retest reliability (Test 2), WOSI-PL questionnaires were filled out by 71 subjects (95.95%) again after 48–72 h (three subjects [4.05%] refused to participate in Test 2). Fifty-one patients (68.92%) who had undergone physiotherapy ($\bar{x}$ = 5.52 ± 1.15; 2–7 weeks) and completed Test 3 were used to determine responsiveness of the questionnaire. Twenty-three patients (31.08%) were excluded: 15 patients (20.27%) did not undergo physiotherapy and 8 patients (10.81%) did not return the completed questionnaires.

The absolute values of the WOSI-PL, SF-36 and QuickDASH for Test 1 are presented in Table 2.

### 3.2. Reliability Analysis

#### 3.2.1. Internal Consistency

Internal consistency of the WOSI-PL was calculated based on the data from Test 1 (*n* = 74). Correlations were computed between particular detailed measures and between specific measures and a summary measure. All domains were closely related (from 0.73 to 0.92) and close relations between the domains and the WOSI-PL Total were also noticeable (from 0.85 to 0.97) (Table 3).

A high degree of internal consistency, reflected by Cronbach’s alpha (α = 0.94), was shown for the WOSI-PL Total, with the domain range of 0.89 to 0.95 (Table 4).

#### 3.2.2. Reliability and Measurement Error (Test–Retest)

The value of ICC_2,1_ (*n* = 71) for the WOSI-PL was very high. It ranged from 0.986 for the Sports/recreation/work domain to 0.994 for the Lifestyle domain and 0.996 for the Total score. SEM ranged from 1.82 in the Lifestyle domain to 3.10 for the Sports/recreation/work domain and 1.41 for the Total score. MDC ranged from 5.05 for the Lifestyle domain to 8.60 for the Sports/recreation/work domain and 3.90 for the Total score (Table 4).

### 3.3. Construct Validity

#### Hypotheses Testing

Table 5 presents the construct validity using the SCC for the WOSI-PL and the reference tools. In line with the hypothesis, there were strong correlations between the scores representing the same areas, which suggests that both tools measured a similar construct. Furthermore, there were moderate correlations between the scores representing less convergent areas, relative to the similarity of the construct (18 hypotheses). Two hypotheses formulated a priori were rejected and the relevant figures are marked in bold. The correlations found between the WOSI-PL Total and the Physical Functioning domain as well as PCS of the SF-36 were strong rather than moderate. Sixteen out of 18 hypotheses formulated a priori (89%) were confirmed (underlined). This suggests high construct validity of WOSI-PL.

### 3.4. Responsiveness

#### 3.4.1. Significance of Changes in WOSI-PL between Test 1 and 3, Standard Effect Size (ES) and Standardized Response Mean (SRM)

The findings show a positive change in the Total score and all the domains of the WOSI-PL (*n* = 51). Following the physiotherapy program, over 80% of the patients were found with improvement in the Total score and in each domain; the changes were statistically significant. ES and SRM for the WOSI-PL were also calculated. The Total score and all domains showed a moderate or large degree of responsiveness, reflected by ES and SRM values (Table 6).

#### 3.4.2. Correlation of Changes in WOSI-PL (Test 1–3) with Changes in QuickDASH (Test 1–3)

All correlations concerning changes in the WOSI-PL between Test 1 and 3 for all domains and Total score with changes in QuickDASH were significant and of moderate power (in the range from 0.39 for Sports/recreation/work domain to 0.58 for Total score) (Table 7).

#### 3.4.3. The Minimal Clinically Important Difference (MCID)

Anchor-Based Method

The MCID for the Total score of the WOSI-PL was 126.43 points (6%), with a 95 per cent confidence interval of 67.83–185.03.

Distribution-Based Method

The MCID for the Total score of the WOSI-PL was 174.05 points (8%) with a 95% confidence interval of 138.61–233.98.

## 4. Discussion

Adaptation of measurement tools allows the global exchange of results in a standardized manner, thereby enabling reliable international comparisons to be made, i.e., assessing the treatment strategies used and the impact of selected factors on them. The WOSI is one of most frequently used questionnaires for shoulder instability with very good psychometric properties [6,8]. It has been validated in twelve languages and in fourteen cultures so far [9,10,11,12,13,14,15,16,17,18,19,20,21,22,23,24,25]. Bejer et al., (2019) published the results of the translation and cultural adaptation of the WOSI into Polish [26].

To the best of our knowledge, our research project was the first attempt to assess the psychometric properties of the Polish version of the WOSI. The majority of the hypotheses specified in the methodology were proven. The findings show that the psychometric properties of the WOSI-PL correspond to those reported for the original version of the tool and WOSI adaptations developed in other countries [3,9,10,11,12,13,14,15,16,17,18,19,20,21,22,23,24,25]. A summary of the psychometric properties of all language versions of the WOSI questionnaire is shown in Appendix A. The evidence presented in the current study supports the validity of the Polish version of the questionnaire as a highly reliable and responsive tool enabling assessment of the quality of life in individuals after arthroscopic repair for shoulder instability.

In terms of reliability of the WOSI-PL, both internal consistency and reproducibility were assessed. The Cronbach’s alpha coefficient for the WOSI-PL was found to be 0.94 (*n* = 74), which demonstrates good coherence between the different questions and indicates an excellent internal consistency. It also exceeds 0.90, which is the recommended threshold when a questionnaire is used in a clinical setting [40]; not exceeding 0.95 which may indicate item redundancy [34]. The Cronbach’s alpha coefficient in other language versions of the WOSI ranged from 0.84 (*n* = 85) in the Japanese version [12] to 0.97 (*n* = 81) in the European Portuguese versions [25]. The original research of the WOSI does not report this coefficient [3].

Positive ratings for test–retest reliability can be reported if ICC is ≥0.70 [34], so the present findings, showing ICC values of 0.99, also reflect good repeatability (interval 48–72 h) of the WOSI-PL (*n* = 71). These values are similar to those reported in the case of the original WOSI (ICC at 2 weeks was 0.95) [3] and are consistent with other studies reporting repeatability of this tool using a similar time interval. In the Turkish version, the ICC was 0.97 [21], and in the European Portuguese version the ICC was 0.97 [25]. In both of these cases, the time interval was 72 h [21,25]. In studies with a longer time interval, the ICC values were slightly lower: the Swedish version [9] (ICC = 0.94; 2 months; *n* = 32), the German version by Drerup et al. [11] (ICC = 0.87; 10 days; *n* = 29), the version for European and North American French-speaking populations by Gaudelli et al. [17] (ICC = 0.87; 6–14 days; *n* = 144) or the French version by Perrin et al. [20] (ICC = 0.88; 7 days; *n* = 27).

We analyzed an SEM and MDC_95_ to assess the error associated with applications of the WOSI-PL. Our findings recorded an SEM of 1.41% and MDC_95_ of 3.90%. This indicates that a patient must change at least 3.9 points (on a scale from 0 to 100) to detect a significant change in shoulder function that can be considered independent of measurement error. For the WOSI-PL domains, the SEM ranged from 1.82 to 3.10%, resulting in MDC_95_ which varied from 5.05 to 8.60%. These results show a slightly lower measurement error than other validations; however, they are still consistent with results of other researchers: Cacchio et al. (SEM of 3.4% and MDC_95_ of 9.3%) [13], Van der Linde et al. (SEM of 8.3% and SDC_95_ of 23%) [16], Wiertsema et al. (SEM of 6.2% and SDC_95_ of 17.2%) [15], Perrin et al. (SEM of 5.7% and MDC_95_ of 15.9%) [20] and Torres et al. (SEM of 3.10% and MDC_95_ of 8.60%) [25]. Yuguero et al. found, in the Spanish version, a greater measurement error than in the other versions (SEM = 23% and MDC = 76%) [18]. Other WOSI validation studies did not calculate SEM or MDC [3,9,10,11,12,14,17,21,22,23].

Construct validity of the WOSI-PL was tested by having eighteen a priori hypotheses. They determined the connections between the WOSI-PL (Total and domain scores), the DASH (Total scores) and the SF-36 (Total and subscale scores). They revealed a stronger association between the WOSI-PL (which is a disease specific PROM) and the DASH (which measures a similar construct but is region specific), but a weaker connection between the WOSI-PL and the SF-36 (which represent a less convergent construct: global health). Sixteen out of 18 a priori assumed hypotheses (89%) were confirmed, which, according to Terwee et al. [34], indicates a high construct validity of the questionnaire. The authors of the original version showed results similar to ours. The WOSI Total score was strongly correlated with the DASH questionnaire (r = 0.77), weaker with the SF-12 Physical Component (r = 0.66) and weakest with the SF-12 Mental Component (r = 0.12). The authors of all published language versions of the WOSI pointed to the correct construct validity of their language versions [9,10,11,12,13,14,16,17,18,19,20,21,22,23,24,25].

To assess responsiveness of WOSI-PL, that is, the questionnaire’s ability to detect clinically important changes over time, distribution-based methods were used (ES and SRM) in the group of patients subjected to physiotherapy after Test 1. According to Husted et al. [38], all domains and the Total score of the WOSI-PL showed a moderate or large degree of responsiveness (Total: ES = 0.44, SRM = 1.26). Only eight previous studies have determined the responsiveness of the WOSI [3,9,13,17,18,22,23,24]. The WOSI SRM value of 0.93 represents a large degree of responsiveness as reported by Kirkley et al. [3]. The researchers demonstrated moderate (SRM = 0.65; ES = 0.62) [41] to large degree of responsiveness (up to SRM = 2.94, ES = 3.17 [24]) in other studies of the WOSI [3,9,13,17,18,22,23,24]. Information on the value of MCID for the WOSI-PL can be applied to determine if the observed changes are meaningful for patients and to estimate the number of patients who achieve a change greater than the MCID following a specific intervention. Our data indicate that the MCID amounts to 126.43 pts. (6% of the 0-2100 WOSI-PL scale) for the anchor-based method, and 174.05 (8% of the 0-2100 WOSI-PL scale) for the distribution-based method. These findings are comparable to the results of studies shown by Kirkley et al. (MCID = 10%) [42] and significantly lower than those reported by Cacchio et al. (MCID = 19%) [13]. The potential reasons for this difference in the findings of the Italian researchers could be related to the fact that a different population of patients was tested and different methods were applied to calculate the MCID.

One limitation of the study is the lack of generalizability of the patients in the study. Only patients who had been treated with arthroscopic repair for shoulder instability were included. The psychometric properties of WOSI-PL therefore may not be generalizable to patients treated conservatively by physiotherapy or by open stabilization. Furthermore, the limited sample size of the female group makes it difficult to perform separate analyses assessing the reliability and validity of the Polish WOSI relative to gender. In line with the literature guidelines followed by our study [34], further research should include a confirmatory factor analysis (CFA), which examines whether the data fit an a priori hypothesized factor structure. We wanted to provide the most accurate sample size for this analysis, since it is recommended that CFA should take into account a group of at least 200–300 subjects [43]. Up until now, factor structure of the language versions of the WOSI was only assessed by researchers from the Netherlands [16] and Spain [18]. The results of the factor analyses (confirmatory factor analyses CFA, exploratory factor analyses EFA) did not allow the researchers to explicitly confirm the validity of the four domains in the relevant language versions of WOSI. The researchers also emphasise it is necessary to continue the related research and analyses.

## 5. Conclusions

The current findings suggest that the Polish version of WOSI is a reliable, valid and responsive patient-reported outcome measure, and it can be recommended for assessing the quality of life in patients following arthroscopic repair for shoulder instability. The WOSI-PL can be used both in related research and in clinical settings, to monitor effects of treatments and to facilitate patient-centred therapeutic decisions.

## Figures and Tables

**Table 1 ijerph-19-14015-t001:** Patient demographic and clinical characteristics (*n* = 74).

	*n* (%)	$\bar{x}$/SD/(Range)
Gender		
Male	64 (86.49)	
Female	10 (13.51)	
Age	(years)	30.01 ± 8.98 (18–50)
Handedness		
Right-handed	68 (91.89)	
Left-handed	6 (8.11)	
Affected side		
Dominant	61 (82.43)	
Non-dominant	13 (17.57)	
Time from arthroscopy	(weeks)	14.42 ± 1.11 (13–16)

Abbreviations: *n* number; $\bar{x}$ Mean; SD standard deviation; % per cent.

**Table 2 ijerph-19-14015-t002:** Absolute values of all scores (*n* = 74).

Questionnaire	Total Group	
	$\bar{x}$ ± SD	Range
**WOSI-PL (%)**		
Physical symptoms	73.8 ± 21.2	26.2–100.0
Sports/recreation/work	63.7 ± 26.7	13.5–100.0
Lifestyle	69.1 ± 24.9	15.3–100.0
Emotions	64.7 ± 26.3	13.0–100.0
Total	69.7 ± 22.5	28.5–99.5
**SF-36 v.2.0**		
Physical Functioning	83.0 ± 12.4	50.0–100.0
Physical Role	66.5 ± 23.3	18.8–100.0
Bodily Pain	66.9 ± 22.2	22.5–100.0
General Health	66.3 ± 15.5	35.0–95.0
Vitality	62.1 ± 17.4	18.8–100.0
Social Functioning	76.7 ± 22.4	12.5–100.0
Emotional Role	82.0 ± 20.5	25.0–100.0
Mental Health	71.8 ± 16.1	20.0–100.0
PCS	74.4 ± 14.0	41.2–97.6
MCS	71.9 ± 16.0	21.4–100.0
**QuickDASH**	21.8 ± 9.3	11.0–47.0

Abbreviations: *n* number; % per cent; $\bar{x}$ Mean; SD standard deviation; WOSI-PL the Western Ontario Shoulder Instability Index, Polish version; QuickDASH the Shortened version of the Disabilities of Arm, Shoulder and Hand Questionnaire; SF-36 v. 2.0 the Short Form-36 version 2.0; PCS the Physical Component Summary; MCS the Mental Component Summary.

**Table 3 ijerph-19-14015-t003:** Spearman’s correlation coefficients (SCC) between the domains of the WOSI-PL (*n* = 74).

WOSI-PL	Physical Symptoms	Sports/Recreation/Work	Lifestyle	Emotions	Total
Physical symptoms	1	0.91	0.91	0.75	0.96
Sports/recreation/ work	0.91	1	0.92	0.73	0.93
Lifestyle	0.91	0.92	1	0.84	0.97
Emotions	0.75	0.73	0.84	1	0.85
Total	0.96	0.93	0.97	0.85	1

Abbreviations: WOSI-PL the Western Ontario Shoulder Instability Index, Polish version. All correlations are statically significant (*p* ≤ 0.001).

**Table 4 ijerph-19-14015-t004:** Results of the reliability analysis: internal consistency (*n* = 74), test–retest reliability and measurement error (*n* = 71).

WOSI-PL	No of Items	Cronbach’s Alpha (α)	ICC_2,1_ (95% CI)	SEM %	MDC_95_%
Physical symptoms	10	0.948	0.988 (0.980–0.993)	2.28	6.31
Sports/recreation/work	4	0.944	0.986 (0.978–0.991)	3.10	8.60
Lifestyle	4	0.920	0.994 (0.991–0.996)	1.82	5.05
Emotions	3	0.899	0.989 (0.982–0.993)	2.72	7.54
Total	21	0.941	0.996 (0.993–0.997)	1.41	3.90

Abbreviations: WOSI-PL the Western Ontario Shoulder Instability Index, Polish version; ICC_2,1_ Intra Class Correlation; CI Confidence Interval; SEM Standard Error of Measurement; MDC Minimal Detectable Change.

**Table 5 ijerph-19-14015-t005:** Correlations (SCC) between the WOSI-PL (domains and total score) and reference questionnaires (*n* = 74).

Questionnaire	WOSI-PL				
	Physical Symptoms	Sports/Recreation/Work	Lifestyle	Emotions	Total
**SF-36**					
Physical Functioning	073	0.72	0.74	0.53	0.72
Physical Role	0.67	0.64	0.63	0.36	0.62
Bodily Pain	0.67	0.68	0.67	0.50	0.67
General Health	0.54	0.53	0.49	0.39	0.51
Vitality	0.19 ^n.s.^	0.22 ^n.s^	0.24	0.37	0.27
Social Functioning	0.35	0.37	0.46	0.44	0.43
Emotional Role	0.43	0.39	0.44	0.37	0.45
Mental Health	0.31	0.32	0.39	0.43	0.39
PCS	0.77	0.75	0.74	0.44	0.74
MCS	0.35	0.33	0.40	0.51	0.42
**QuickDASH**	−0.83	−0.80	−0.78	−0.58	−0.80

Abbreviations: N number; WOSI-PL the Western Ontario Shoulder Instability Index, Polish version; QuickDASH the Shortened version of the Disabilities of Arm, Shoulder and Hand Questionnaire; SF-36 v. 2.0 the Short Form-36 version 2.0; PCS the Physical Component Summary; MCS the Mental Component Summary. All correlations are statically significant (*p* ≤ 0,05); n.s. non-significance.

**Table 6 ijerph-19-14015-t006:** Results of the responsiveness analysis (N = 51).

WOSI-PL	Test 1	Test 3		Results of Physiotherapy		
	$\bar{x}$ ± SD	$\bar{x}$ ± SD	*p*	%_↑_	ES	SRM
Physical symptoms	67.69 ± 21.02	77.79 ± 16.53	*p* < 0.001 *	94.1%	0.44	1.23
Sports/recreation/work	56.89 ± 26.32	67.07 ± 21.12	*p* < 0.001 *	90.2%	0.37	1.02
Lifestyle	61.99 ± 25.35	72.29 ± 20.02	*p* < 0.001 *	84.3%	0.39	0.87
Emotions	59.30 ± 26.02	71.48 ± 21.86	*p* < 0.001 *	88.2%	0.45	0.96
Total	63.35 ± 22,06	73,8 ± 17,83	*p* < 0.001 *	98.0%	0.44	1.26

Abbreviations: N number; WOSI-PL the Western Ontario Shoulder Instability Index, Polish version; SD standard deviation; %**_↑_** percentage of patients with improvement in the WOSI-PL; ES effect size; SRM standardized response mean; p Wilcoxon test for paired samples; * statistically significant (*p* ≤ 0.05).

**Table 7 ijerph-19-14015-t007:** Spearman’s correlation coefficients (SCC) between changes (test 1–3) in all domains and Total score of the WOSI PL and changes in the QuickDASH (*n* = 51).

WOSI–PL (Test 1–3)	QuickDASH (Test 1–3)
	SCC
Physical symptoms	r = −0.544. *p* < 0.001 *
Sports/recreation/work	r = −0.387. *p* = 0.005 *
Lifestyle	r = −0.499. *p* < 0.001 *
Emotions	r = −0.446. *p* = 0.001 *
Total	r = −0.584. *p* < 0.001 *

Abbreviations: N number; WOSI-PL the Western Ontario Shoulder Instability Index, Polish version; QuickDASH the Shortened version of the Disabilities of Arm, Shoulder and Hand Questionnaire; *SCC* Spearman’s correlation coefficients. * statistically significant (*p* ≤ 0.05).

## Data Availability

The data that support the findings of this study are available from the corresponding author upon reasonable request.

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
