# Peer review of "Reliability, Validity and Responsiveness of the Polish Version of the Western Ontario Shoulder Instability Index (WOSI-PL) in Patients after Arthroscopic Repair for Shoulder Instability"

_ijerph, 2022, doi:10.3390/ijerph192114015_

Round 1
Reviewer 1 Report
Article, Title: Reliability, validity and responsiveness of the Polish version of the Western Ontario Shoulder Instability Index (WOSI-PL) in patients after arthroscopic repair for shoulder instability The aim of this study was to evaluate psychometric properties of the Polish version of Western Ontario Shoulder Instability Index (WOSI) - one of the most frequently used patient-reported outcome measures regarding SI. The Polish version of WOSI can be considered as a reliable, valid and responsive PROMs, recommended for assessing the quality of life in patients after arthroscopic repair for SI. It could be applied in research and clinical setting for monitoring treatment, and facilitating patient-centred therapeutic decisions.
Reviewer 2 Report
The paper presents the psychometric study of the Polish version of the Western Ontario Shoulder Instability Index (WOSI), which is a valued contribution to the Polish population with shoulder instability and the work adds to the international evidence of the WOSI psychometric properties. This is a timely and important topic, as upper limb function highly impacts everyday performance and participation, and so it is necessary to have reliable and valid instruments.
The manuscript is well-written, well-presented and the procedures for the psychometric validation. I only have one major issue and some minor issues for the authors to clarify.
Major issue
1. The authors explore the internal consistency, test-retest reliability and the concurrent ("construct") validity of the WOSI in Polish population. However, they do not explore the factorial validity, which may be very interesting, specially to test if the structure of the Polish WOSI remains the same as of the original and other cross-culturally adapted versions. They do briefly mention this aspect in the Limitation section, but I would like the authors to expand on this topic: for instance, they could clarify wether the original WOSI structure has been confirmed in other cross-culturally adapted versions.
Minor issues
Introduction
2. The authors presents the topic briefly but provide enough information to understand the state of the art. They mention several PROMs aimed to measure shoulder instability and offer enough rationale to go for the WOSI instead of the rest. However, while the authors mentions that the WOSI has been adapted to several languages, they do not explain the Polish cross-cultural adaptation in the Introduction. They need to move the information of lines 293-296 to the Introduction, so the reader knows there is a previously adapted version of the WOSI into Polish population that has met the standars of such process.
Materials and Methods
3. There is a typo in line 90 ("Patients___18 years").
4. The authors need to provide more information regarding the comparison measures (i.e., QuickDASH, SF-36 and GRC), specifically regarding wehter these instruments are adapted, reliable and valid in the Polish population, as the authors are using them to test the construct validity of the WOSI.
5. I would recommend to not refer to the Tables in the Discussion (e.g., "Table 8" in lines 323, 336, 349, 359 and 368).
6. The male-female ratio is extremmely unbalanced, and this is not discussed nor analyzed within the work. As the limited sample size of the female group makes it difficult to perform separate analyses to confirm the reliability and validity of the Polish WOSI according sex, the authors need to state this as a major limitation of the study.
Reviewer 3 Report
Dear Authors,
I have a few small suggestions, you can find them in the pdf I have uploaded. However, in the plagiarism report, the rate was 31 percent. Although it seems that the author mostly plagiarized from his previous articles, this situation is not acceptable for the published journal. Authors should review the article to minimize plagiarism.

Reviewer 4 Report
Consistent and standardized diagnostic methods are important for objective assessment of patients' conditions and monitoring of treatment and rehabilitation. The work corresponds well with this task. The primary objective was to validate the Polish version of the well-known functional assessment scale of the shoulder joint. The validation process was carried out correctly according to the accepted rules. The structure of the work corresponds to the requirements for this type of work. The work does not contain factual and technical errors. It can be submitted for publication without corrections.
Round 2
Reviewer 3 Report
Dear Authors,
I think your revisions are appropriate. However, plagiarism appears to be conspicuous, especially from this source (Bejer, A., Ćwirlej-SozaÅ„ska, A., WiÅ›niowska-Szurlej, A., Wilmowska-PietruszyÅ„ska, A., Spalek, R., De Sire, A., SozaÅ„ski, B., 2021. Psychometric properties of the Polish version of the 36-item WHODAS 2.0 in patients with hip and knee osteoarthritis. Quality of Life Research 30, 2415–2427.. doi:10.1007/s11136-021-02806-4). Authors are required to make minor revisions taking into account the plagiarism report.
Best wishes
